# Use of Larval Morphological Deformities in *Chironomus plumosus* (Chironomidae: Diptera) as an Indicator of Freshwater Environmental Contamination (Lake Trasimeno, Italy)

**Enzo Goretti** [1,*], **Matteo Pallottini** [1], **Sarah Pagliarini** [1], **Marianna Catasti** [1],
**Gianandrea La Porta** [1], **Roberta Selvaggi** [1], **Elda Gaino** [1], **Alessandro Maria Di Giulio** [2]
**and Arshad Ali** [3]

1   Dipartimento di Chimica, Biologia e Biotecnologie, Università degli Studi di Perugia, 06123 Perugia, Italy;
    matteo.pallottini@unipg.it (M.P.); sarah.pagliarini87@gmail.com (S.P.); marianna.catasti@gmail.com (M.C.);
    gianandrea.laporta@unipg.it (G.L.P.); roberta.selvaggi@unipg.it (R.S.); elda.gaino@unipg.it (E.G.)
2   Servizio Disinfestazione, USLUmbria1, 06127 Perugia, Italy; alessandro.digiulio@uslumbria1.it
3   Entomology and Nematology Department, University of Florida, Gainesville, FL 32611, USA; umar@ufl.edu
*   Correspondence: enzo.goretti@unipg.it

**Abstract:** The mentum deformity incidence in *Chironomus plumosus* larvae to assess the environmental contamination level in Lake Trasimeno, Central Italy, was investigated. The survey lasted from May 2018 to August 2019. Fifty-one samplings were carried out: 34 in the littoral zone and 17 in the central zone. The deformity assessment was based on 737 and 2767 larval specimens of *C. plumosus* collected from the littoral and central zones, respectively. Comparison of the larval morphometric variables between normal and deformed specimens highlighted that the deformities did not cause alterations of the larval growth. The deformity incidence amounted to 7.22% in the whole Trasimeno's ecosystem, reaching 8.28% in the littoral zone and 6.94% in the central zone. Among the different seasonal cohorts, the spring cohort had overall the highest deformity value (11.41%). The deformity type assessment protocol highlighted that the most common deformity type was "round/filed teeth" (64%). The results of this 2018–2019 survey revealed a low deformity incidence, within the background range of relatively low-impacted freshwaters. Comparison with previous investigations (2000–2010) of the same habitat showed a clear decrease of the deformity incidence. This study further contributes to the evaluation of the mentum deformity in chironomids that represent an indicator endpoint of the anthropogenic contamination level in freshwaters.

**Keywords:** Chironomidae; *Chironomus plumosus* larvae; mentum deformities; freshwater contamination; Lake Trasimeno

---

## 1. Introduction

Anthropic activities continuously add urban, industrial, and agricultural wastes into the environment. Common and widespread groups of pollutants, such as heavy metals, pesticides, polycyclic aromatic hydrocarbons, and polychlorinated biphenyls are dispersed into the environment forming a complex mixture whose toxic effects are difficult to quantify [1–3].

Normally, the set of contaminants contributing to the total toxic burden is not well known and, even when the exact composition is known, it is difficult to interpret the estimate of potential effects of simple additivity or synergism or antagonism [4]. In fact, the measurement of concentrations of toxic elements or compounds in the environment does not imply knowledge of the resulting toxicity of their

mixture on the biota [5]. Furthermore, the peculiar trophic network of living organisms [6] determines bioaccumulation and biomagnification processes of contaminants [7–9].

For an evaluation of the anthropogenic contamination in a given territory, the biological monitoring of freshwaters seems to be the most appropriate strategy. The rains draining the atmosphere and the soil carry pollutants into the water receptor of the underlying hydrographic basin [10,11].

Among aquatic organisms, benthic macroinvertebrates inhabiting the surficial layer of the bottom sediments in inland waters are valid biological indicators because they are subject to the resulting action of pollutants and thus respond to many ecological stressors [12–16]. Aquatic Diptera typically represent the predominant component of the benthic biocoenosis, and among them chironomids (Chironomidae: Diptera) in particular are characterized by diverse species compositions and have a key role in the trophic network of freshwater ecosystems especially those impaired by substantial organic loads [17]. Therefore, chironomid communities represent an important food source for a wide range of aquatic and terrestrial animals [18], also considering in some cases, the role of pestiferous chironomid species [19].

Some chironomid species are frequently used as endpoints (as test organisms) in ecotoxicological bioassays for the assessment of ecological risks; for example, larval growth rate, adult emergence, and survival [20]. In particular, widely used in bioassays are the larvae of aquatic midge species belonging to the genus *Chironomus*, such as *Chironomus plumosus* (Linnaeus, 1758), *Chironomus riparius* (Meigen, 1804), *Chironomus tentans* (Fabricius, 1805), *Chironomus sancticaroli* (Strixino and Strixino, 1981), and *Chironomus tepperi* (Skuse, 1889) [21–25].

In freshwater ecosystems, these bioindicator chironomid species are common, abundant, and tolerant to polluted sediments. The exposure to potentially contaminated sediments is due to their benthic habits and feeding on fine particulate organic matter (POM: 0.5 μm–1 mm) collected from sediments [18]. Thus, they exhibit suitability in monitoring the pollutant effects in inland waters.

In several laboratory and field studies, an association between environmental contamination level (metals, such as As, Cd, Cr, Cu, Hg, Mn, Ni, Pb, Zn, and organic compounds, such as polychlorinated biphenyls (PCBs) and pesticides) and the onset of sublethal effects as deformities of mouthparts (i.e., mentum, mandibles, and pectin epipharyngis) in chironomid larvae has been observed. In this regard, studies on chironomid mouthpart deformities have been developed since the 1970s [26–28] and a vast amount of literature based on laboratory bioassays [22,23,29–41] and field surveys [21,42–47] endorses that the incidence of these deformities is well associated with the degree of sediment toxicity, whereas no relationship has been detected between deformed phenotypes and organic enrichment; hence, the water quality assessment indices are not always consistent with the use of chironomid deformities [48]. However, some studies have expressed concerns about the association between deformity incidence and toxicants [24,49].

The mouthpart deformities probably are a consequence of the interaction of contaminants, playing as endocrine-disrupting chemicals, with hormones structurally related to estrogen, such as ecdysone, that leads and regulates insect growth and metamorphosis [50]. Therefore, the "altered" ecdysone might interfere during the molting processes in larval development, causing phenotypic anomalies, as the mouthpart deformities in chironomids [51].

Thus, morphological deformities in chironomids are an aspecific response to different contaminants, such as heavy metals and pesticides, and an ecotoxicological endpoint to evaluate the total effects of the exposure to a toxic mixture of peculiar toxicants present in the freshwater environments [4,22]. For this reason, chironomid deformities are a reliable biomarker for assessing environmental toxicity.

The purpose of the present research was: (i) to use the incidence of mentum deformities in *C. plumosus* larvae as an effective tool for biological assessment of the environmental contamination entity in Lake Trasimeno, Central Italy; (ii) to compare the results with previous investigation in Lake Trasimeno (nearly ten years ago [43]) and evaluate the enhancement of environmental contamination over time.

## 2. Materials and Methods

### 2.1. Study Area

Lake Trasimeno, the fourth largest lake in Italy, is located within the Tiber River basin in Umbria, Central Italy. It is the most extended lake of peninsular Italy (124.3 km² surface). It is a shallow lake of tectonic origin, at hydrometric zero (257.33 meters above sea level (m a.s.l.)) and has an average depth of 4.7 m with a maximum depth of 6.3 m [52,53]. It is characterized by a Mediterranean climate, and its hydrological regime depends on local precipitations, which shows considerable fluctuations in water level over the years [54,55]. These peculiar features and the diffuse anthropogenic sources of pollution (agriculture, livestock farms, industry, and urban settlements) in the lake surrounding land areas make this biotope susceptible to contamination [56]. For its naturalistic and ecological importance, Lake Trasimeno is considered as a Special Area of Conservation (IT5210018; Habitat Directive, 1992/43/EEC) and Special Protection Area (IT5210070; Bird Directive, 2009/147/EC), and it is also a regional park.

### 2.2. Field Sampling Campaign

The field sampling campaign was carried out in the littoral and central zones of the lake. In the littoral zone, six sampling sites were selected along a 350 m area, representative of the main bottom types (sandy, silty, and macrophyte-covered) present in Lake Trasimeno, situated along the coastline (1.5 m average depth) close to the town of Castiglione del Lago. This area among the littoral zone is most subject to the anthropic impact on the lake. In the central zone of the lake, three sampling sites were selected in the same bathymetric zone (4–5 m depth) (Figure 1). The sampling design in the present study was based on a previous investigation [43] carried out from 2000 to 2010 at Lake Trasimeno.

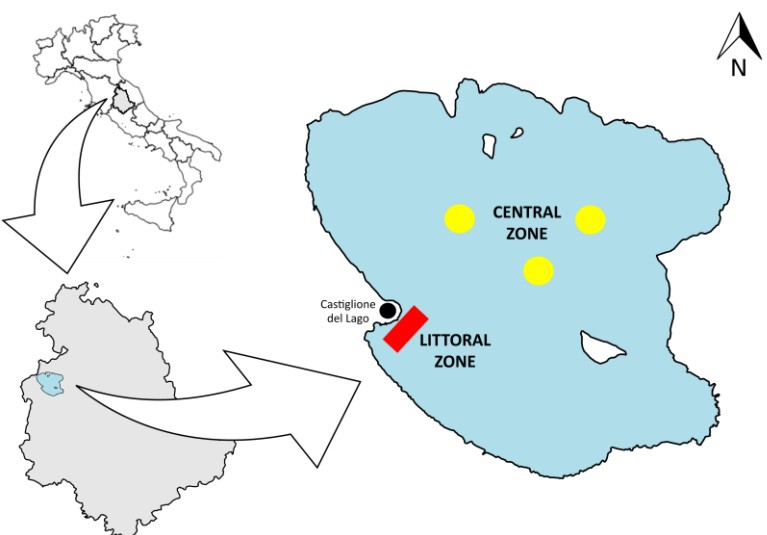

**Figure 1.** Lake Trasimeno, Umbria, Italy, and location of the field sampling sites.

The survey lasted for 16 months: from May 2018 to August 2019. A total of 51 samplings were carried out: 34 in the littoral zone and 17 in the central zone (Table 1). For each of the two zones, at least one monthly survey was conducted (with the exception of June 2018 for the central zone), with a higher sampling frequency during the spring–summer period.

During each sampling occasion, in the littoral zone 6, quantitative sub-samples of the bottom sediments were collected at each sampling site (a total of 36 sub-samples) with a hand dredge (112 cm² sampling surface); in the central zone the sampling technique consisted of a quantitative sampling of the bottom sediments with a compressed air dredge (400 cm² sampling surface, 21 mesh per cm nylon net), repeated 5 times per site, with a duration of 1 min each (a total of 15 sub-samples).

**Table 1.** *Chironomus plumosus* individuals (N), *C. plumosus* relative percentages within the chironomid community (%), Chironomidae densities (individuals m$^{-2}$), Chironomidae relative percentages within the total macroinvertebrates (%), Naididae densities (individuals m$^{-2}$), Naididae relative percentages within the total macroinvertebrates (%), other taxa densities (individuals m$^{-2}$), other taxa relative percentages within the total macroinvertebrates (%), macroinvertebrate densities (individuals m$^{-2}$); minimum (Min), median, mean, standard deviation (SD), and maximum (Max) of these parameters; (**a**) littoral zone, (**b**) central zone.

| | | | | **(a) Littoral Zone** | | | | | |
|---|---|---|---|---|---|---|---|---|---|
| **Date** | *C. plumosus* | *C. plumosus* | Chironomidae | Chironomidae | Naididae | Naididae | Other Taxa | Other Taxa | Macroinvertebrates |
| | **(N)** | **(% of Chironomidae)** | **(ind. m$^{-2}$)** | **(%)** | **(ind. m$^{-2}$)** | **(%)** | **(ind. m$^{-2}$)** | **(%)** | **(ind. m$^{-2}$)** |
| 11/05/2018 | 2 | 2.94 | 183 | 9.24 | 1793 | 90.40 | 7 | 0.36 | 1983 |
| 18/05/2018 | 6 | 4.44 | 469 | 28.08 | 1188 | 71.17 | 12 | 0.74 | 1669 |
| 25/05/2018 | 4 | 2.26 | 646 | 35.67 | 1154 | 63.68 | 12 | 0.65 | 1812 |
| 01/06/2018 | 16 | 11.85 | 518 | 35.13 | 935 | 63.36 | 22 | 1.51 | 1476 |
| 14/06/2018 | 7 | 3.87 | 599 | 42.62 | 578 | 41.12 | 228 | 16.26 | 1405 |
| 22/06/2018 | 2 | 1.60 | 426 | 39.29 | 536 | 49.41 | 122 | 11.29 | 1084 |
| 28/06/2018 | 6 | 8.33 | 239 | 23.76 | 706 | 70.23 | 60 | 6.01 | 1006 |
| 11/07/2018 | 16 | 23.53 | 242 | 23.29 | 670 | 64.56 | 126 | 12.15 | 1037 |
| 19/07/2018 | 12 | 29.27 | 122 | 9.51 | 1099 | 86.02 | 57 | 4.47 | 1277 |
| 25/07/2018 | 8 | 19.05 | 152 | 14.32 | 814 | 76.98 | 92 | 8.70 | 1058 |
| 01/08/2018 | 35 | 51.47 | 242 | 18.11 | 993 | 74.41 | 100 | 7.48 | 1334 |
| 09/08/2018 | 28 | 20.90 | 452 | 26.55 | 1182 | 69.48 | 68 | 3.97 | 1702 |
| 22/08/2018 | 21 | 24.42 | 376 | 20.10 | 1451 | 77.51 | 45 | 2.39 | 1872 |
| 03/09/2018 | 72 | 46.15 | 490 | 27.51 | 1250 | 70.20 | 41 | 2.29 | 1781 |
| 18/09/2018 | 102 | 76.69 | 439 | 35.76 | 743 | 60.60 | 45 | 3.64 | 1226 |
| 27/09/2018 | 12 | 37.50 | 179 | 13.77 | 1103 | 85.02 | 16 | 1.21 | 1297 |
| 10/10/2018 | 15 | 12.93 | 567 | 25.65 | 1641 | 74.23 | 3 | 0.12 | 2211 |
| 17/10/2018 | 33 | 13.25 | 1079 | 34.09 | 2066 | 65.27 | 20 | 0.64 | 3166 |
| 24/10/2018 | 75 | 35.55 | 843 | 47.95 | 903 | 51.34 | 12 | 0.71 | 1758 |
| 06/11/2018 | 8 | 6.11 | 457 | 48.29 | 490 | 51.71 | 0 | 0 | 947 |
| 04/12/2018 | 9 | 6.47 | 472 | 58.18 | 339 | 41.82 | 0 | 0 | 811 |
| 16/01/2019 | 3 | 1.75 | 533 | 53.00 | 462 | 45.95 | 11 | 1.04 | 1006 |
| 12/02/2019 | 16 | 10.53 | 505 | 51.70 | 472 | 48.30 | 0 | 0.00 | 977 |
| 15/03/2019 | 0 | 0 | 258 | 34.90 | 469 | 63.42 | 12 | 1.68 | 739 |
| 04/04/2019 | 6 | 4.17 | 463 | 32.93 | 944 | 67.07 | 0 | 0 | 1407 |
| 02/05/2019 | 13 | 3.75 | 1008 | 41.73 | 1375 | 56.91 | 33 | 1.36 | 2415 |
| 21/05/2019 | 7 | 7.95 | 304 | 25.70 | 829 | 70.19 | 48 | 4.10 | 1181 |
| 04/06/2019 | 16 | 7.69 | 722 | 32.59 | 1409 | 63.61 | 84 | 3.81 | 2215 |
| 18/06/2019 | 33 | 17.19 | 623 | 38.20 | 952 | 58.45 | 55 | 3.35 | 1629 |
| 27/06/2019 | 11 | 14.47 | 252 | 42.67 | 328 | 55.56 | 11 | 1.78 | 591 |
| 09/07/2019 | 68 | 83.95 | 262 | 48.66 | 262 | 48.66 | 14 | 2.67 | 539 |
| 23/07/2019 | 24 | 55.81 | 130 | 13.97 | 760 | 81.64 | 41 | 4.38 | 931 |

**Table 1.** *Cont.*

| (a) Littoral Zone | | | | | | | | | |
|---|---|---|---|---|---|---|---|---|---|
| **Date** | *C. plumosus* | *C. plumosus* | Chironomidae | Chironomidae | Naididae | Naididae | Other Taxa | Other Taxa | Macroinvertebrates |
| | **(N)** | **(% of Chironomidae)** | **(ind. m$^{-2}$)** | **(%)** | **(ind. m$^{-2}$)** | **(%)** | **(ind. m$^{-2}$)** | **(%)** | **(ind. m$^{-2}$)** |
| 06/08/2019 | 72 | 40.22 | 656 | 24.74 | 1885 | 71.13 | 110 | 4.14 | 2651 |
| 22/08/2019 | 38 | 17.33 | 880 | 15.13 | 4655 | 80.01 | 283 | 4.86 | 5818 |
| Min | 0 | 0.00 | 122 | 9.24 | 262 | 41.12 | 0 | 0.00 | 539 |
| Median | 14 | 13.09 | 460 | 32.76 | 939 | 64.91 | 37 | 2.34 | 1369 |
| Mean | 23 | 20.69 | 464 | 31.55 | 1072 | 64.98 | 53 | 3.46 | 1589 |
| SD | 25 | 21.41 | 246 | 13.17 | 782 | 12.90 | 64 | 3.83 | 955 |
| Max | 102 | 83.95 | 1079 | 58.18 | 4655 | 90.40 | 283 | 16.26 | 5818 |
| (b) Central Zone | | | | | | | | | |
| **Date** | *C. plumosus* | *C. plumosus* | Chironomidae | Chironomidae | Naididae | Naididae | Other Taxa | Other Taxa | Macroinvertebrates |
| | **(N)** | **(% of Chironomidae)** | **(ind. m$^{-2}$)** | **(%)** | **(ind. m$^{-2}$)** | **(%)** | **(ind. m$^{-2}$)** | **(%)** | **(ind. m$^{-2}$)** |
| 29/05/2018 | 107 | 81.68 | 252 | 16.76 | 1250 | 83.24 | 0 | 0 | 1502 |
| 18/07/2018 | 396 | 98.02 | 685 | 75.69 | 220 | 24.31 | 0 | 0 | 905 |
| 08/08/2018 | 274 | 90.43 | 577 | 50.96 | 548 | 48.45 | 7 | 0.59 | 1132 |
| 22/08/2018 | 274 | 89.54 | 542 | 55.75 | 430 | 44.25 | 0 | 0 | 972 |
| 21/09/2018 | 117 | 100 | 202 | 16.40 | 1028 | 83.60 | 0 | 0 | 1230 |
| 16/10/2018 | 68 | 100 | 120 | 21.24 | 445 | 78.76 | 0 | 0 | 565 |
| 28/11/2018 | 66 | 100 | 113 | 12.88 | 767 | 87.12 | 0 | 0 | 880 |
| 20/12/2018 | 107 | 98.17 | 208 | 22.12 | 732 | 77.70 | 2 | 0.18 | 942 |
| 31/01/2019 | 82 | 100 | 143 | 22.34 | 498 | 77.66 | 0 | 0 | 642 |
| 21/02/2019 | 61 | 96.83 | 112 | 27.57 | 293 | 72.43 | 0 | 0 | 405 |
| 08/03/2019 | 86 | 100 | 145 | 27.44 | 383 | 72.56 | 0 | 0 | 528 |
| 12/04/2019 | 38 | 97.44 | 65 | 25.49 | 190 | 74.51 | 0 | 0 | 255 |
| 17/05/2019 | 69 | 100 | 123 | 28.57 | 308 | 71.43 | 0 | 0 | 432 |
| 13/06/2019 | 78 | 100 | 145 | 21.22 | 538 | 78.78 | 0 | 0 | 683 |
| 11/07/2019 | 74 | 98.67 | 137 | 37.61 | 225 | 61.93 | 2 | 0.46 | 363 |
| 31/07/2019 | 598 | 99.50 | 1115 | 37.54 | 1853 | 62.40 | 2 | 0 | 2970 |
| 08/08/2019 | 567 | 100 | 1008 | 43.94 | 1287 | 56.06 | 0 | 0 | 2295 |
| Min | 38 | 81.68 | 65 | 12.88 | 190 | 24.31 | 0 | 0.00 | 255 |
| Median | 86 | 99.50 | 145 | 27.44 | 498 | 72.56 | 0 | 0.00 | 880 |
| Mean | 180 | 97.07 | 335 | 31.97 | 647 | 67.95 | 1 | 0.08 | 982 |
| SD | 180 | 5.11 | 330 | 16.57 | 461 | 16.62 | 2 | 0.18 | 715 |
| Max | 598 | 100.00 | 1115 | 75.69 | 1853 | 87.12 | 7 | 0.59 | 2970 |

Benthic macroinvertebrates were separated from the sediment, sorted, and then preserved in 70% ethanol for later examination in the laboratory. The main taxa composing the benthic community of each sample were firstly identified to the family level using suitable taxonomic keys [57]. Chironomid specimens were subsequently identified to genus or species (*Chironomus plumosus*) level through the observation of peculiar features of their head capsules [58–61].

### 2.3. Mouthpart Deformities of Chironomus plumosus Larvae

Body length and head capsule dimensions (width and length) of chironomid larvae were measured to identify the larval stage (instar), and head capsules were then mounted on slides and fixed with Faure liquid (modified composition [22,43]).

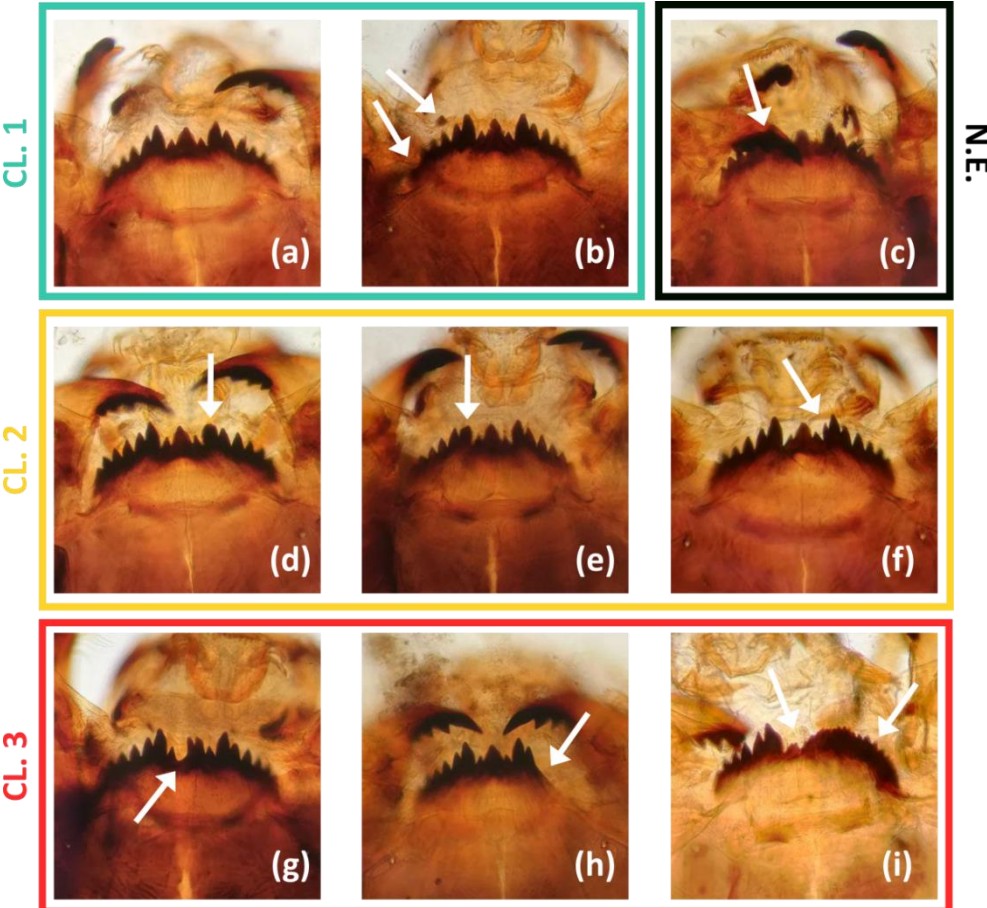

**Figure 2.** *Chironomus plumosus* images showing Class 1 (CL. 1): (**a**) the normal-shaped mentum, (**b**) the normal-shaped mentum with mechanical damage (arrows), (**c**) the mentum not evaluable (N.E., not completely visible, arrow), and the various mentum deformity types of Class 2 (CL. 2) and Class 3 (CL. 3). (**d**) CL. 2: mentum with a rounded tooth (arrow), (**e**) mentum with a filed tooth (arrow), and (**f**) mentum with an additional tooth (arrow). (**g**) CL. 3: mentum with a Köhn gap (arrow), (**h**) mentum with missing teeth (arrow), and (**i**) mentum with a combination of different deformities (arrows).

Only *Chironomus plumosus* head capsules were considered for mouthpart deformity assessment and examined under a microscope. Specimens that did not have completely visible mentum or those with a cephalic capsule highly damaged during fixation on the microscope slides were excluded from further analyses. *Chironomus plumosus* specimens were classified according to the protocol for morphological response of the mentum, consisting of three classes [22,42–44,62]: Class 1 (CL. 1) specimens without morphological deformity or with mechanical damages (broken or worn menta or teeth) [4,5,63–65]; Class 2 (CL. 2) specimens with weak deformity (one or two round/filed teeth; one

missing or additional tooth; one bifid tooth; one serrate tooth; two joined teeth; weak asymmetry); Class 3 (CL. 3) specimens with severe deformity (very round/filed teeth; two or more missing or additional teeth; two or more bifid teeth; serrate teeth; three or more joined teeth; severe asymmetry; Köhn gap; combination of different deformities). Examples of mentum deformity classes of *C. plumosus* larvae are shown in Figure 2.

### 2.4. Statistical Analysis

Ljung–Box test was used to analyze temporal autocorrelation. Generalized linear models (GLM) were used to analyze differences in head width and length among the different deformity classes, setting larval instar as a covariate and Gaussian family as the link function. The test of equal proportions was carried out to examine differences in the deformity incidences among the seasons [66]. Each deformity class was tested independently and the pairwise comparisons for proportions with the Bonferroni's correction were used as a post-hoc procedure. The results were regarded as significant at a two-tailed *p*-value of 0.05. All calculations were performed using the glm function and the prop.test of the R software [67].

## 3. Results

### 3.1. Macroinvertebrates and Chironomids

During this survey at Lake Trasimeno, a total of 31,240 benthic macroinvertebrate specimens were collected: 21,220 in the littoral zone and 10,020 in the central zone. The benthic communities were mainly composed of Oligochaeta (Naididae) and Diptera: (Chironomidae). The littoral zone community was composed of the following means (±SD): Naididae: 64.98% (±12.90%), Chironomidae: 31.55% (±13.17%), and other taxa: 3.46% (±3.83%). A very similar benthic community composition was found in the central zone: Naididae: 67.95% (±16.62%), Chironomidae: 31.97% (±16.57%), and other taxa: 0.08% (±0.18%). The other macroinvertebrate taxa collected were essentially alien invasive species (i.e., *Dreissena polymorpha* (Pallas, 1771) and *Dikerogammarus villosus* (Sowinsky, 1894)) (Table 1).

During the whole sampling period, the mean (±SD) density of Chironomidae was 464 (±246) and 335 (±330) individuals $m^{-2}$, in the littoral and central zones, respectively; the mean (±SD) density of Naididae was 1072 (±782) and 647 (±461) individuals $m^{-2}$, in the littoral and central zones, respectively; and the mean (±SD) density of other taxa was 53 (±64) and 1 (±2) individuals $m^{-2}$, in the littoral and central zones, respectively (Table 1).

A total of 3857 *C. plumosus* larvae was collected, 796 from the littoral zone and 3061 from the central zone. In the littoral zone, *C. plumosus* represented a mean (±SD) of 20.69% (±21.41%) of the total chironomid community (ranging from 0% to 83.95%), while it was always the dominant taxon of the central zone chironomid community (mean 97.07% ± 5.11%), ranging from 81.68% to 100% (Table 1).

The analysis of biometric variables identified the respective instar of each specimen. In the littoral zone most (60.93%) specimens belonged to instar IV, 29.27% to instar III, and 9.80% to instar II. Almost all specimens (99.25%) from the central zone belonged to instar IV, while only 0.72% and 0.03% belonged to instar III and II, respectively.

### 3.2. Mentum Deformity in Chironomus plumosus

For mentum deformity evaluation of *C. plumosus* larvae, 59 specimens collected from the littoral zone and 294 collected from the central zone of Lake Trasimeno were excluded due to damages to the cephalic capsule or due to incomplete visibility of mentum. Thus, deformity assessment was based on 737 specimens from the littoral zone and on 2767 specimens from the central zone. Ljung–Box test revealed no temporal autocorrelation within deformity classes among samplings. The deformity incidence amounted to 7.22% (1.51% in CL. 3) in the whole Lake Trasimeno's ecosystem, reaching 8.28% (3.12% in CL. 3) in the littoral zone and 6.94% (1.08% in CL. 3) in the central zone of the lake (Table 2).

**Table 2.** *Chironomus plumosus* incidence of the deformity classes: CL. 1, no deformity; CL. 2, weak deformity; CL. 3, strong deformity and CL. (2 + 3); individuals (N) and percentages (%); (**a**) littoral zone; (**b**) central zone; (**c**) Lake Trasimeno.

| (a) Littoral Zone | | | (b) Central Zone | | | (c) Lake Trasimeno | | |
|---|---|---|---|---|---|---|---|---|
| CLASS | N | % | CLASS | N | % | CLASS | N | % |
| CL. 1 | 676 | 91.72 | CL. 1 | 2575 | 93.1 | CL. 1 | 3251 | 92.78 |
| CL. 2 | 38 | 5.16 | CL. 2 | 162 | 5.85 | CL. 2 | 200 | 5.71 |
| CL. 3 | 23 | 3.12 | CL. 3 | 30 | 1.08 | CL. 3 | 53 | 1.51 |
| CL. (2 + 3) | 61 | 8.28 | CL. (2 + 3) | 192 | 6.94 | CL. (2 + 3) | 253 | 7.22 |

The polyvoltine biological cycle of *C. plumosus* at Lake Trasimeno exhibited one winter cohort, one spring cohort, and different summer cohorts during the year. Thus, the mentum deformity classes were analyzed in the different seasonal cohorts as shown in a box-plot of the relative statistical variables in Figure 3. In particular, in the littoral zone (34 samplings), the deformed specimens (CL. 2 + 3) had mean values of 1.0 ± 0, 5.0 ± 3.6, and 10 ± 3.0 in the winter, spring, and summer cohorts, respectively; while in the central zone (17 samplings) the deformed specimens (CL. 2 + 3) had mean values of 4.5 ± 1.5, 9.3 ± 7.5, and 45.7 ± 33.1 in the winter, spring, and summer cohorts, respectively.

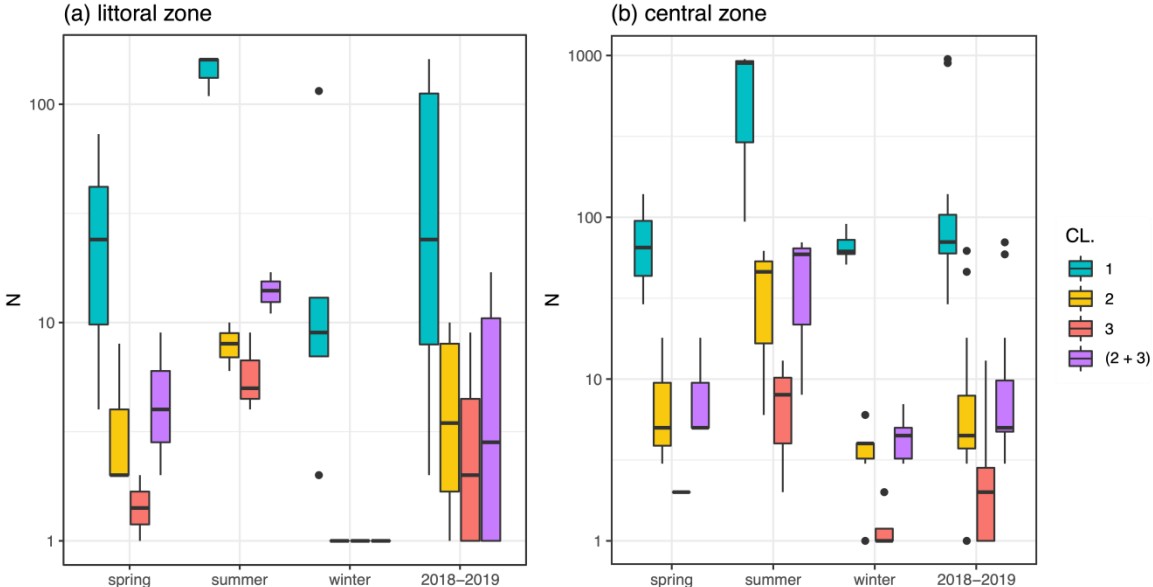

**Figure 3.** Boxplot of *C. plumosus* specimens (N, log scale) in seasonal cohorts and overall 2018–2019 survey, grouped by deformity classes: CL. 1 (green, no deformity), CL. 2 (yellow, weak deformity), CL. 3 (red, strong deformity), and CL. (2 + 3) (purple). (**a**) Littoral zone; (**b**) central zone. Box represents the interquartile range, thick lines represent the median, whiskers represent the minimum and maximum values, and dots represent the outliers.

In the littoral zone the deformity incidence of the winter cohort was 2.67% (1.33% in CL. 3), the spring cohort was 12.93% (2.59% in CL. 3), and the summer cohort was 8.92% (3.82% in CL. 3); while in the central zone the deformities in the winter cohort amounted to 6.32% (1.17% in CL. 3), 10.73% in the spring cohort (0.77% in CL. 3), and 6.59% in the summer cohort (1.11% in CL. 3) (Figure 4).

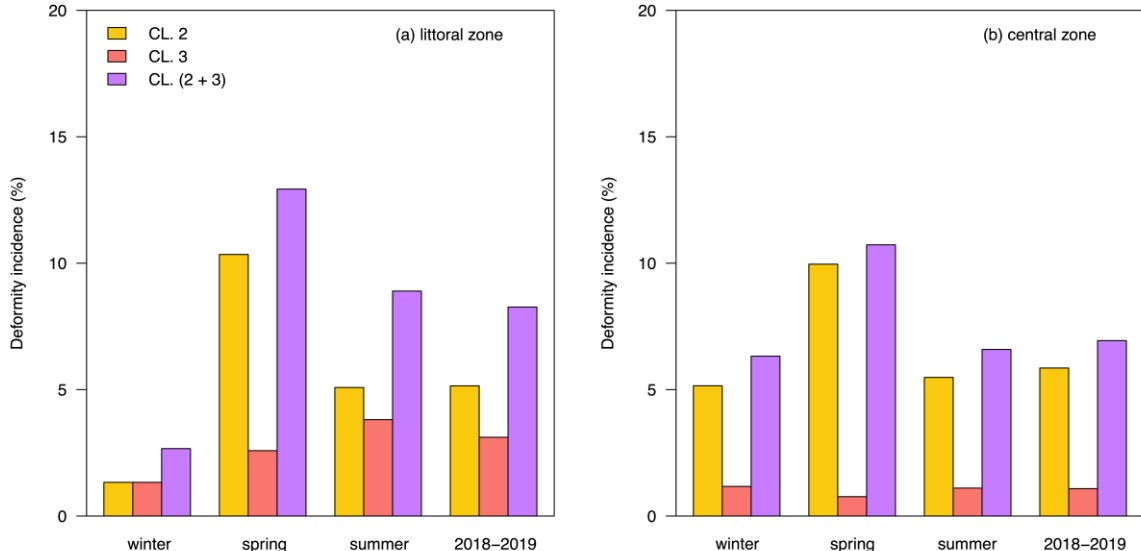

**Figure 4.** Deformity incidence (%): CL. 2 (yellow, weak deformity); CL. 3 (red, strong deformity); and CL. (2 + 3) (purple) in *C. plumosus* seasonal cohorts and overall 2018–2019 survey; (**a**) littoral zone; (**b**) central zone.

In the littoral zone, the test of equal proportions showed significant differences in the rate of deformity incidence among the seasons for CL. 2 ($p < 0.01$). In contrast, no differences were observed for CL. 3 ($p > 0.6$). Combining CL. 2 and CL. 3, significant differences were detected with an increase in the proportion of specimens with deformities in spring ($p < 0.01$). In the central zone, significant differences were observed for CL. 2, with a deformity incidence in spring significantly higher than that recorded in summer ($p < 0.01$), while no differences were detected for CL. 3 ($p > 0.9$). Combining CL. 2 and CL. 3, no statistical differences emerged.

The deformity assessment in relation to the different larval stages (instars) highlighted that in the littoral zone 7.33% of the deformed specimens (2.85% in CL. 3) belonged to instar IV and only 0.95% (0.27% in CL. 3) to instar III, while in the central zone the instar IV deformed specimens were 6.90% (1.08% in CL. 3) and only 0.04% (none in CL. 3) belonged to instar III. No deformed specimens in both zones belonged to instar II (Table 3).

Figure 5 shows the distribution and relative head size (head length and width) of normal and deformed larvae among instars, in the littoral and central zones of Lake Trasimeno. In the littoral zone, the head length and width means of instar IV specimens (about 61% of total *C. plumosus* larvae) belonging to CL. 1 (normal), CL. 2 (weak deformity), and CL. 3 (strong deformity) were 0.93 and 0.73 mm, 0.93 and 0.73 mm, and 0.95 and 0.75 mm, respectively. Similar trends of head length and width of instar IV specimens (about 99% of total *C. plumosus* larvae) were recorded in the central zone, the mean sizes were 0.98 and 0.78 mm (CL. 1), 0.98 and 0.78 mm (CL. 2), and 1.00 and 0.80 mm (CL. 3) (Table 3). No significant differences were observed in the biometric variables of the head among the three deformity classes ($p > 0.05$).

**Table 3.** Head length and width (mm) of *C. plumosus* larvae (all instars, instar IV, instar III, and instar II); CL. 1, no deformity; CL. 2, weak deformity; CL. 3, strong deformity; and CL. (2 + 3); number (N), minimum (Min), median, mean, standard deviation (SD), and maximum (Max); (**a**) littoral zone; (**b**) central zone.

| | | (a) Littoral Zone | | | | | | | | (b) Central Zone | | | | | | | |
|---|---|---|---|---|---|---|---|---|---|---|---|---|---|---|---|---|---|
| | | Head Lenght (mm) | | | | Head Width (mm) | | | | Head Lenght (mm) | | | | Head Width (mm) | | | |
| CL | | 1 | 2 | 3 | 2 + 3 | 1 | 2 | 3 | 2 + 3 | 1 | 2 | 3 | 2 + 3 | 1 | 2 | 3 | 2 + 3 |
| All Instars | N | 676 | 38 | 23 | 61 | 676 | 38 | 23 | 61 | 2575 | 162 | 30 | 192 | 2575 | 162 | 30 | 192 |
| | Min | 0.18 | 0.48 | 0.40 | 0.40 | 0.13 | 0.38 | 0.38 | 0.38 | 0.30 | 0.53 | 0.85 | 0.53 | 0.25 | 0.43 | 0.70 | 0.43 |
| | Median | 0.83 | 0.91 | 0.95 | 0.93 | 0.65 | 0.75 | 0.75 | 0.75 | 1.00 | 1.00 | 1.00 | 1.00 | 0.78 | 0.78 | 0.79 | 0.78 |
| | Mean | 0.75 | 0.88 | 0.90 | 0.89 | 0.58 | 0.69 | 0.72 | 0.70 | 0.98 | 0.98 | 1.00 | 0.98 | 0.78 | 0.78 | 0.80 | 0.78 |
| | SD | 0.24 | 0.17 | 0.16 | 0.17 | 0.20 | 0.13 | 0.12 | 0.13 | 0.09 | 0.10 | 0.09 | 0.10 | 0.08 | 0.08 | 0.08 | 0.08 |
| | Max | 1.20 | 1.10 | 1.10 | 1.10 | 0.93 | 0.95 | 0.90 | 0.95 | 1.25 | 1.25 | 1.25 | 1.25 | 1.00 | 1.05 | 1.00 | 1.05 |
| Instar IV | N | 402 | 33 | 21 | 54 | 402 | 33 | 21 | 54 | 2555 | 161 | 30 | 191 | 2555 | 161 | 30 | 191 |
| | Min | 0.70 | 0.75 | 0.75 | 0.75 | 0.50 | 0.60 | 0.63 | 0.60 | 0.70 | 0.68 | 0.85 | 0.68 | 0.50 | 0.55 | 0.70 | 0.55 |
| | Median | 0.93 | 0.93 | 0.95 | 0.95 | 0.75 | 0.75 | 0.75 | 0.75 | 1.00 | 1.00 | 1.00 | 1.00 | 0.78 | 0.78 | 0.79 | 0.78 |
| | Mean | 0.93 | 0.93 | 0.95 | 0.94 | 0.73 | 0.73 | 0.75 | 0.74 | 0.98 | 0.98 | 1.00 | 0.98 | 0.78 | 0.78 | 0.80 | 0.79 |
| | SD | 0.10 | 0.09 | 0.08 | 0.08 | 0.08 | 0.07 | 0.07 | 0.07 | 0.08 | 0.09 | 0.09 | 0.09 | 0.07 | 0.07 | 0.08 | 0.07 |
| | Max | 1.20 | 1.10 | 1.10 | 1.10 | 0.93 | 0.95 | 0.90 | 0.95 | 1.25 | 1.25 | 1.25 | 1.25 | 1.00 | 1.05 | 1.00 | 1.05 |
| Instar III | N | 214 | 5 | 2 | 7 | 214 | 5 | 2 | 7 | 19 | 1 | - | 1 | 19 | 1 | - | 1 |
| | Min | 0.40 | 0.48 | 0.40 | 0.40 | 0.20 | 0.38 | 0.38 | 0.38 | 0.50 | 0.53 | - | 0.53 | 0.30 | 0.43 | - | 0.43 |
| | Median | 0.53 | 0.50 | 0.45 | 0.50 | 0.40 | 0.40 | 0.39 | 0.40 | 0.55 | 0.53 | - | 0.53 | 0.40 | 0.43 | - | 0.43 |
| | Mean | 0.53 | 0.51 | 0.45 | 0.49 | 0.40 | 0.40 | 0.39 | 0.40 | 0.55 | 0.53 | - | 0.53 | 0.41 | 0.43 | - | 0.43 |
| | SD | 0.05 | 0.03 | 0.07 | 0.05 | 0.05 | 0.03 | 0.02 | 0.03 | 0.04 | - | - | - | 0.04 | - | - | - |
| | Max | 0.70 | 0.55 | 0.50 | 0.55 | 0.50 | 0.45 | 0.40 | 0.45 | 0.63 | 0.53 | - | 0.53 | 0.45 | 0.43 | - | 0.43 |
| Instar II | N | 60 | - | - | - | 60 | - | - | - | 1 | - | - | - | 1 | - | - | - |
| | Min | 0.18 | - | - | - | 0.13 | - | - | - | 0.30 | - | - | - | 0.25 | - | - | - |
| | Median | 0.33 | - | - | - | 0.23 | - | - | - | 0.30 | - | - | - | 0.25 | - | - | - |
| | Mean | 0.31 | - | - | - | 0.22 | - | - | - | 0.30 | - | - | - | 0.25 | - | - | - |
| | SD | 0.05 | - | - | - | 0.04 | - | - | - | - | - | - | - | - | - | - | - |
| | Max | 0.40 | - | - | - | 0.33 | - | - | - | 0.30 | - | - | - | 0.25 | - | - | - |

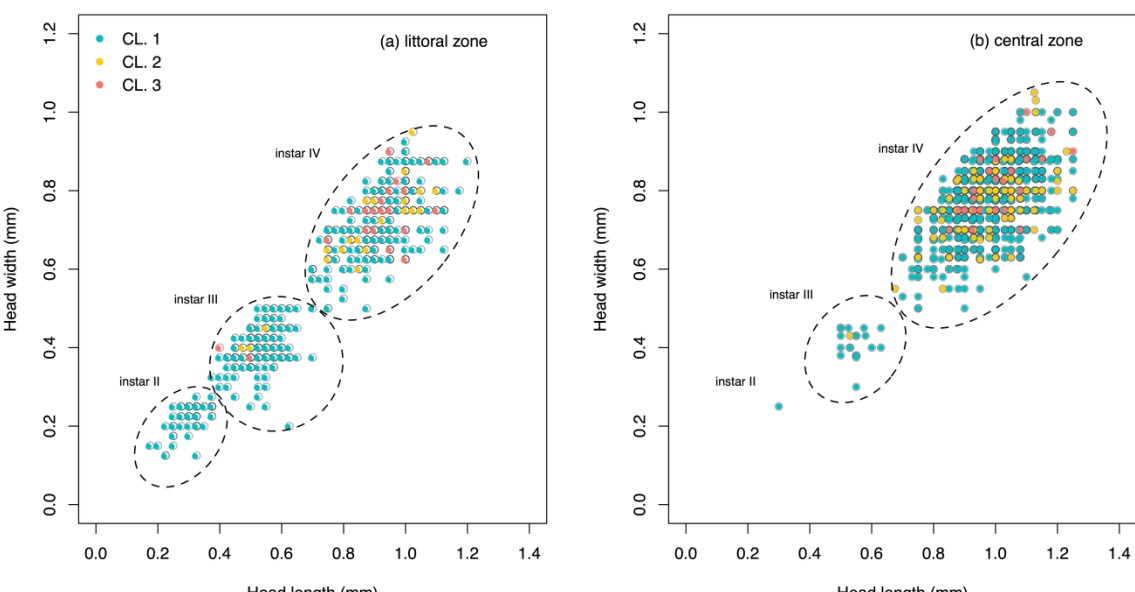

**Figure 5.** Size (length and width; mm) of *C. plumosus* larvae, different instars (II, III, and IV) are surrounded by dotted circles, mentum in CL. 1 (green, no deformity), CL. 2 (yellow, weak deformity), and CL. 3 (red, strong deformity). (**a**) littoral zone; (**b**) central zone.

Lastly, in Table 4 the different types of *C. plumosus* mentum deformities in Lake Trasimeno are reported. The main CL. 2 deformity types in the littoral zone, central zone, and the whole lake, respectively, were: "one or two round/filed teeth" with 81.58%, 69.75%, and 72.00%; "one missing tooth" with 2.63%, 17.90%, and 15.00%; "one additional tooth" with 5.26%, 3.09%, and 3.50%; "one

serrate tooth" with 10.53%, 3.70%, and 5.00%; "one bifid tooth" with 0%, 4.94%, and 4.00%. The main CL. 3 deformity types, listed in the same order, were: "very round/filed teeth" with 39.13%, 33.33%, and 35.85%; "two or more missing teeth" with 21.74%, 16.67%, and 18.87%; "two or more additional teeth" with 8.70%, 0%, and 3.77%; "Köhn gap" with 0%, 20.00%, and 11.32%.

**Table 4.** *Chironomus plumosus* mentum deformity types: CL. 2 (weak deformity), CL. 3 (strong deformity), and CL. (2 + 3); numbers (N) and percentages (%); (**a**) littoral zone; (**b**) central zone; (**c**) Lake Trasimeno, Central Italy.

| Class and Deformity Type | (a) Littoral Zone | | (b) Central Zone | | (c) Lake Trasimeno | |
|---|---|---|---|---|---|---|
| | (N) | (%) | (N) | (%) | (N) | (%) |
| **CL. 2** | **38** | | **162** | | **200** | |
| One or Two Round/Filed Teeth | 31 | 81.58 | 113 | 69.75 | 144 | 72.00 |
| One Missing Tooth | 1 | 2.63 | 29 | 17.90 | 30 | 15.00 |
| One Additional Tooth | 2 | 5.26 | 5 | 3.09 | 7 | 3.50 |
| One Bifid Tooth | 0 | 0 | 8 | 4.94 | 8 | 4.00 |
| One Serrate Tooth | 4 | 10.53 | 6 | 3.70 | 10 | 5.00 |
| Two Joined Teeth | 0 | 0 | 1 | 0.62 | 1 | 0.50 |
| Weak Asymmetry | 0 | 0 | 0 | 0 | 0 | 0 |
| **CL. 3** | **23** | | **30** | | **53** | |
| Very Round/Filed Teeth | 9 | 39.13 | 10 | 33.33 | 19 | 35.85 |
| Two or More Missing Teeth | 5 | 21.74 | 5 | 16.67 | 10 | 18.87 |
| Two or More Additional Teeth | 2 | 8.70 | 0 | 0 | 2 | 3.77 |
| Two or More Bifid Teeth | 0 | 0 | 0 | 0 | 0 | 0 |
| Serrate Teeth | 0 | 0 | 0 | 0 | 0 | 0 |
| Three or More Joined Teeth | 0 | 0 | 0 | 0 | 0 | 0 |
| Severe Asymmetry | 0 | 0 | 0 | 0 | 0 | 0 |
| Köhn Gap | 0 | 0 | 6 | 20.00 | 6 | 11.32 |
| Combination of Different Deformities | 7 | 30.43 | 9 | 30.00 | 16 | 30.19 |
| **CL. (2 + 3)** | **61** | | **192** | | **253** | |
| Round/Filed Teeth | 40 | 65.57 | 123 | 64.06 | 163 | 64.43 |
| Missing Teeth | 6 | 9.84 | 34 | 17.71 | 40 | 15.81 |
| Additional Teeth | 4 | 6.56 | 5 | 2.60 | 9 | 3.56 |
| Bifid Teeth | 0 | 0 | 8 | 4.17 | 8 | 3.16 |
| Serrate Teeth | 4 | 6.56 | 6 | 3.13 | 10 | 3.95 |
| Joined Teeth | 0 | 0 | 1 | 0.52 | 1 | 0.40 |
| Asymmetry | 0 | 0 | 0 | 0 | 0 | 0 |
| Köhn Gap | 0 | 0 | 6 | 3.13 | 6 | 2.37 |
| Combination of Different Deformities | 7 | 11.48 | 9 | 4.69 | 16 | 6.32 |

## 4. Discussion

This study analyzed mentum deformities in *C. plumosus* larvae as an endpoint for evaluating the effects of anthropogenic stressors in the shallow Lake Trasimeno. The results of the survey showed a low deformity incidence, amounting to about 7% (1.5% in CL. 3) in the whole Lake Trasimeno's ecosystem, with similar values in the littoral and central zones of the lake, about 8% (3% in CL. 3) and 7% (1% in CL. 3), respectively.

Paleolimnological studies, representing natural background levels during the pre-industrial age, showed deformity incidences between 0% and 0.8% [68,69]. In the contemporary age [70,71] the incidence of background deformities in low-impacted areas is lower than 8% suggesting this threshold as the reference background for deformity incidence in natural inland waters. Therefore, the results of the present research affirm that this biotope has a deformity incidence value comparable to that of inland waters subject to low anthropic impact. Perhaps, this can be associated to a low degree of contamination of the Trasimeno's ecosystem, in particular of sediments that represent a preferential sink for toxic compounds and, at the same time, serve as a habitat for chironomid larvae.

The deformity incidence analysis among the different cohorts of the polyvoltine biological cycle of *C. plumosus* showed that the spring cohort had overall the highest deformity values in Lake Trasimeno, corresponding to 11.41% (1.33% in CL. 3). Perhaps, this peak of deformities in the spring cohort is due to the compromise between metabolic activity and residence time in sediments, compared to the winter cohort (low metabolic activity and long residence time) and to summer cohorts (high metabolic activity and short residence time).

Head size (length and width) of normal and deformed larvae grouped according to their instar showed similar trends, regardless of whether they were found in the littoral or in the central zone of the lake. These results highlight that mentum deformity did not cause alterations of the head size of deformed specimens, meaning a similar larval growth of the latter compared to normal specimens [62].

A comparison of current results with previous investigations (2000–2010) conducted in Lake Trasimeno using the same sampling methods showed a clear decrease in the incidence of mentum deformities in *C. plumosus* larvae, both in the littoral and in the central zone of the lake [43]. In fact, the deformity incidences that were about 20% (9.5% in CL. 3) in the littoral zone of Castiglione del Lago during 2006–2009 and about 13% (3.8% in CL. 3) in the central zone of the lake during 2000–2010, decreased to 8% and 7% in the respective lake areas during the present survey (2018–2019). This 10-year comparison demonstrated a clear decrease in the current deformity index values. Over the years, in effect, there has been a confirmation of this decreasing trend of deformity incidence that, in the littoral zone, decreased from 31% in 2006 to 10% in 2009, until reaching the current level of 8%. A similar phenomenon tended to occur in the central zone of the lake as well, which showed a deformity index of 11% in 2000, 17% in 2001, 10% in 2002, and 13% in October 2010, until reaching the current level of 7%.

In summary, the surveys carried out at Lake Trasimeno allow a space–time biological assessment of lacustrine sediments' contamination showing a qualitative recovery over time of the lake ecosystem both in the central and littoral zone; the latter zone, more easily subject to the introduction of pollutants, showed an incidence of severe deformities (CL. 3) of 3% compared to 1% in the central zone.

However, the relationship between the incidence of strong (CL. 3) and weak (CL. 2) deformities is not easy to interpret, in fact, even if severe deformed phenotypes (CL. 3) were found in specimens subject to sediments contaminated both with organic and inorganic pollutants, their increase is probably influenced not only by the quantity but also by the type of compounds that form the toxic mixture [4,22]. Consequently, as a reliable biomarker, it is more appropriate to consider the two deformity classes together (CL. 2 + 3) for an overall assessment of the toxic effects of sediment contamination [22].

In this regard, some studies based on ecotoxicological laboratory experiments queried the relationship between chironomid deformities and toxicity assessment in freshwater ecosystems [24,49,72–76]. However, the high variability of those experimental results can be influenced by different variables, such as: (i) the endpoint used (mentum, mandibles, pectin epipharyngis); (ii) the deformity types considered; (iii) the larval instars used in the experiment; (iv) the type of cultures used which often cause a high incidence of deformities in the control test; (v) the exposure time of the larvae to the contaminants; and (vi) the type of contaminant mixture used in the test.

On the other hand, in natural environments the association between incidence of chironomid deformities and levels of chemical contamination is very consistent, despite the lack of a standardized protocol for the deformity assignment which can generate cases of inconsistent interpretation. In this respect, in the ring test (based on digital images) [74] involving 25 international experts to assess the status (normal or deformed) of 211 chironomid larvae menta (IV instar) from Lake Saimaa (Finland), the majority of them (at least 2/3) highlighted clearly that the incidence of mentum deformities was high in the polluted site (about 30%) and low in the reference site (<10%), with no significant differences between blind or open assessment.

In addition, the present study contributes to the elaboration of a detailed protocol for the definition of which types of mentum deformities to be considered for an assessment of the deformity incidence in *Chironomus* populations of freshwater biotopes.

The present study data showed that, among deformed menta of *C. plumosus* found in Lake Trasimeno (littoral and central zones), the most common deformity types were "round/filed teeth" with 64% followed by "missing teeth" (16%), a "combination of different deformities" (about 6%), "serrate teeth" (4%), "additional teeth" (4%), "bifid teeth" (3%), "Köhn gap" (2%), and "joined teeth" (0.4%) (Table 4).

In conclusion, the present study on Lake Trasimeno provides a further contribution to the evaluation of mentum deformities in chironomid populations of inland waters. In the biomonitoring programs, the deformity incidence can provide an easy biological tool to assess the effects of sediment-associated toxicants. The mentum deformities assessment is complementary and non-substitutive of the chemical instrumental analyses; in fact, a high incidence of deformities in a freshwater ecosystem is a warning (indicator) for the start of a chemical analytical monitoring program aimed at qualitative and quantitative detection of the toxic compounds causing environmental contamination.

**Author Contributions:** Conceptualization, E.G. (Enzo Goretti), M.P., S.P., M.C., G.L.P., E.G. (Elda Gaino), and A.A.; Data curation, E.G. (Enzo Goretti), M.P., S.P., M.C., and G.L.P.; Methodology, E.G. (Enzo Goretti), M.P., S.P., M.C., G.L.P., R.S., and A.M.D.G.; Investigation, E.G. (Enzo Goretti), M.P., S.P., M.C., G.L.P., R.S., and A.M.D.G.; Writing—original draft, E.G. (Enzo Goretti), M.P., G.L.P., E.G. (Elda Gaino), and A.A.; Writing—review and editing, E.G. (Enzo Goretti), M.P., G.L.P., E.G. (Elda Gaino), and A.A. All authors have read and agreed to the published version of the manuscript.

**Funding:** This work was supported by the "Fondazione Brunello e Federica Cucinelli" within the 2017–2020 research project "Chironomid population control at Lake Trasimeno: evaluation of biological control methods and new systems for the attraction and mechanical-light capture".

**Acknowledgments:** We thank Andrea Pagnotta (amateur fisherman), Marco Chiappini (Provincia di Perugia), Alberto Fais, and Francesco Giglietti (USL Umbria1), Rosalba Padula and Luca Nicoletti (ARPAUmbria) for their great support in the sampling campaign.

**Conflicts of Interest:** The authors declare no conflicts of interest.

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
