# Peer review of "Use of Larval Morphological Deformities in Chironomus plumosus (Chironomidae: Diptera) as an Indicator of Freshwater Environmental Contamination (Lake Trasimeno, Italy)"

_water, doi:10.3390/w12010001_

Round 1
Reviewer 1 Report
Overall the study is a classical meristic based approach to describing a population over time. The sampling effort is noted and the design seems well put together. The focus on linking deformation to environmental pollution is not support, however, as there was no direct linkage made to incorporate environmental data to chironomid development in the introduction, methods or results. I would suggest a through revision of the intro and discussion to focus more directly on chironomid morphological variation (i.e. head deformation) with a broader focus on ecological and genetic hypotheses to explain the findings put forth. There also needs to be some clarification regarding the analyses and a more robust model that incorporates variation in time and sampling zone.
Abstract: Provide an explanation as to why deformations were expected. Were there any factors that attributed or related to the rate of deformation across samples? The concluding sentence is not supported by the preceding text.
31-82: paragraphs need to be merged or extended. There is a general lack of background linking deformations to pollutant activity.
41-47: revise for clarity
48-53: chironomids should be described in more detail, particularly with regards to the intended study objectives
54-60: provide more details regarding the exact relationships between chrionomids and environmental pollutants
64-74: provide clearer support for the study hypotheses, particularly linking chironomid deformation to environmental pollution. At present there is no support to suggest that finding deformation among samples is anything related to pollution. There are also effects of plasticity and simple interpolation morphological variation that need to be discussed.
118-128: How many deformations are likely a result of mounting or degradation of the individuals during capture or storage? Using a measure that is based on the loss of particular features is a bit risky when handling preserved specimens. At present the groupings seem more qualitative than quantitative, which makes differentiating groups rather subjective. Consider including images to support the different classification groups proposed. Isn’t degradation of teeth a normal part of a creatures lifecycle?
130-136: What error distribution was used? How was temporal autocorrelation dealt with? Was sampling location considered?
132-136: This could be implemented in the glm or via a mixed effects model that would allow a more complete test.
139-158: Provide the standard deviations for mean statistics
Table 2: Are these percent deformations per instar classes or number of individuals per instar? I assume the latter and they legend needs to be corrected. I would suggest checking that the inclusion of the class 1 group is not affecting the statistics by running the analyses without the class and the class on its own.
Figure 3: I would suggest replacing the bar char with box plots that show standard error for each set of groups. Regardless error bars should be included as they will help explain the statistic results.
258-260: I do not see how the results support this conclusion. The test were on the qualitative deformation class assignment differences among instar classes and among seasons. I would expect the results to be more related to differences in development among lower instars, possibly due to different feeding strategies used by the low instars.
247-321: The general focus on pollution is not supported by the available data and needs to be revised.
Author Response
Overall the study is a classical meristic based approach to describing a population over time. The sampling effort is noted and the design seems well put together. The focus on linking deformation to environmental pollution is not support, however, as there was no direct linkage made to incorporate environmental data to chironomid development in the introduction, methods or results. |
I would suggest a through revision of the intro and discussion to focus more directly on chironomid morphological variation (i.e. head deformation) with a broader focus on ecological and genetic hypotheses to explain the findings put forth. There also needs to be some clarification regarding the analyses and a more robust model that incorporates variation in time and sampling zone.
In accordance with your recommendations we made a review of the various sections of the manuscript (from Abstract to Discussion) to better clear the objectives, methods and results of the research project, so that the experimental model could be more robust and solid. The variations made on the different sections are reported in the following points.
Abstract: Provide an explanation as to why deformations were expected.
In this regard, we added the following sentence in the abstract: "The comparison with previous investigations (2000-2010) showed a clear decrease of the deformity incidence", this now provides a reference of the entity of the deformed specimens and a motivation of the study for a comparison after 10 years.
Abstract: Were there any factors that attributed or related to the rate of deformation across samples?
The focus of this work was not the research of the causes of morphological deformities of Chironomid larvae (and therefore they have not been investigated) but it was the definition of a protocol for the identification of the deformity types of the mentum, in order to assess the deformity incidence as an effective tool for the biological assessment of freshwater contamination.
However, it is known from scientific literature (previous contributions also from our laboratory) that, in natural environments, the individual contaminants are of difficult association with the magnitude of the deformity incidence, that is instead influenced by the mixture of toxicants which originate a peculiar toxic mixture.
Abstract: The concluding sentence is not supported by the preceding text.
The rewriting of the abstract solved this problematic aspect.
31-82: paragraphs need to be merged or extended. There is a general lack of background linking deformations to pollutant activity.
In accordance with your suggestions we reformulated the main parts of the “Introduction” section.
41-47: revise for clarity
We re-wrote these sentences.
48-53: chironomids should be described in more detail, particularly with regards to the intended study objectives
We integrated the text with some information on the Diptera chironomids in regard of the objectives of the study, in particular about the availability, tolerance and lifestyle of these organisms in inland water.
54-60: provide more details regarding the exact relationships between chrionomids and environmental pollutants
In this regard, we integrated the text with a sentence on the use of some species of Chironomus in ecotoxicological bioassays as test organisms for the ecological risk evaluation.
64-74: provide clearer support for the study hypotheses, particularly linking chironomid deformation to environmental pollution. At present there is no support to suggest that finding deformation among samples is anything related to pollution. There are also effects of plasticity and simple interpolation morphological variation that need to be discussed.
We reformulated the text trying to clarify the study hypotheses.
118-128: How many deformations are likely a result of mounting or degradation of the individuals during capture or storage? Using a measure that is based on the loss of particular features is a bit risky when handling preserved specimens. At present the groupings seem more qualitative than quantitative, which makes differentiating groups rather subjective. Consider including images to support the different classification groups proposed. Isn’t degradation of teeth a normal part of a creatures lifecycle?
During the mounting of the samples on the slides the most frequent alteration is the breakage of the mentum, an easily identifiable alteration that normally does not prejudice the analysis of the "true" deformities.
Furthermore, in order to avoid doubts about subjective identifications, we took into consideration the suggestion to insert a synoptic image that summarizes and clarifies the different deformity cases analyzed.
130-136: What error distribution was used? How was temporal autocorrelation dealt with? Was sampling location considered?
132-136: This could be implemented in the glm or via a mixed effects model that would allow a more complete test.
Taking into account your suggestions we integrated our statistical analyses on the database. In particular, the evaluation of the error distribution with the GLM test and the analysis of the temporal autocorrelation of the samples has been inserted.
Regarding the Equal proportions tests used to examine differences in the deformity incidences among the seasons [60] we believe that it is a tough test, and it is specific for proportions, and also it is currently used in our laboratory. For this reason, we thought to maintain this statistical method.
139-158: Provide the standard deviations for mean statistics
In accordance with your suggestion we have included in the text of the manuscript the standard deviations for mean statistics.
Table 2: Are these percent deformations per instar classes or number of individuals per instar? I assume the latter and they legend needs to be corrected. I would suggest checking that the inclusion of the class 1 group is not affecting the statistics by running the analyses without the class and the class on its own.
In Table 2 are reported the deformity percentages for number of individuals of Chironomid larvae. The magnitude of the deformities for number of individuals per instar is reported in Table 3. These values refer to the littoral and central zones, and to the lake ecosystem as a whole.
Figure 3: I would suggest replacing the bar char with box plots that show standard error for each set of groups. Regardless error bars should be included as they will help explain the statistic results.
In agreement with the suggestion we included a "box plot" (Figure 2 _R1) that shows the main statistical parameters for each grouping of deformity class in their respective seasonal cohorts.
258-260: I do not see how the results support this conclusion. The test were on the qualitative deformation class assignment differences among instar classes and among seasons. I would expect the results to be more related to differences in development among lower instars, possibly due to different feeding strategies used by the low instars.
In accordance with the suggestions we reformulated the sentence.
247-321: The general focus on pollution is not supported by the available data and needs to be revised.
According to the suggestions we rewrote the Discussion section so that the statements are always supported by the results of the study.
Finally, we revised the english text
Reviewer 2 Report
General comments:
The theme of the article is interesting and is one of the emerging trends in environmental studies.
The introductory remarks justify the theme and are adequately presented.
The sampling campaign is well described in the methods and is suitable for the aim of the study.
The aims of the study are clearly stated in the introduction but the last one is not really carried out.
The statistical analysis are described adeqautely in the methods and the results are generally well presented.
The findings are not new in the literature but they can be relevant for the field of exotoxicology.
The abstract can be improved by removing some methodological details and adding some results about the comparisons over time.
I only have few minor comments.
Detailed comments:
Page 2, lines 46-47: This sentence ca be removed.
Page 2, line 48: Add "benthic" to "macroinvertebrates".
Page 2, line 50: Substitute "all" with "many".
Page 2, lines 55-57: Authors must be between brackets.
Page 2, line 63: Please provide a relevant citation.
Page 2, lines 81-82: Delete the last aims or address it in the paper. "iii) to assess how mentum deformities represent a toxicity indicator in freshwater ecosystems".
Page 3, line 112: State here the identification level and provide the relevant citations.
Page 6, line 184-185: Add confidence interval to figure 2 or switch to box-plot.
Add the information about the results of the test of equal proportion in figure 2 so it will be possible to visually see what is statistically different and what is not.
Page 11, line 285: Substitute "survey" with "surveys".
Page 12, line 307: Substitute "majority them" with "majority of them".
Page 12, lines 310-314: It is not clear how this study contribute to the elaboration of the protocol nor how those information per-se (i.e. without information about the contamination level) can contribute to establish a link between deformities and contamination.
Author Response
General comments:
The theme of the article is interesting and is one of the emerging trends in environmental studies.
The introductory remarks justify the theme and are adequately presented.
The sampling campaign is well described in the methods and is suitable for the aim of the study.
The aims of the study are clearly stated in the introduction but the last one is not really carried out.
According to your suggestions we deleted the point iii), this objective, actually, was not supported from the study results.
The statistical analysis are described adeqautely in the methods and the results are generally well presented. The findings are not new in the literature but they can be relevant for the field of exotoxicology.
The abstract can be improved by removing some methodological details and adding some results about the comparisons over time.
I only have few minor comments.
In accordance with the suggestions, we rewrote the abstract by integrating the text with the main results of the study, among those that were not reported in the previous version (max 200 words allowed).
Detailed comments:
Page 2, lines 46-47: This sentence ca be removed.
In accordance with the suggestions we deleted the sentence.
Page 2, line 48: Add "benthic" to "macroinvertebrates".
We added the term “benthic” to “macroinvertebrates”.
Page 2, line 50: Substitute "all" with "many".
We replaced “all” with “many”.
Page 2, lines 55-57: Authors must be between brackets.
Only for the authors who have described species that changed their genus over the years (following systematic reviews) is it foreseen the insertion between brackets
Page 2, line 63: Please provide a relevant citation.
The sentence was inserted in a different context and therefore it is now supported by relevant citations
Page 2, lines 81-82: Delete the last aims or address it in the paper. "iii) to assess how mentum deformities represent a toxicity indicator in freshwater ecosystems".
See point 1): in agreement with the suggestions we deleted the point iii) of the objectives, actually not supported by the results of the study.
Page 3, line 112: State here the identification level and provide the relevant citations.
In this regard, we included the level of identification of taxa (families, genus and species) and the relative references to the analytical key of freshwater macroinvertebrates used (Tachet et al., 2010)
Page 6, line 184-185: Add confidence interval to figure 2 or switch to box-plot.
Add the information about the results of the test of equal proportion in figure 2 so it will be possible to visually see what is statistically different and what is not.
In accordance with the suggestion we inserted a "box plot" (Figure 2_R1) that shows the main statistical parameters for each group of deformity class in their respective seasonal cohorts
Page 11, line 285: Substitute "survey" with "surveys".
We replaced “surveys” with “surveys”.
Page 12, line 307: Substitute "majority them" with "majority of them".
We replaced “majority them” with “majority of them”.
Page 12, lines 310-314: It is not clear how this study contribute to the elaboration of the protocol nor how those information per-se (i.e. without information about the contamination level) can contribute to establish a link between deformities and contamination.
In agreement with your suggestion we rewrote this sentence to clarify that the contribution to the elaboration of the protocol is based on “which types of mentum deformities to consider for an assessment of the deformity incidence in Chironomus populations of freshwater biotopes".
Finally, we revised the English text
Reviewer 3 Report
This research investigation is about the evaluation of the freshwater Lake Trasimeno, Italy. The authors collected several samples of macroinvertebrates in the central area and in the margin during more than 1 year. The deformity found in the Chironomus plumosus mentum was about 7% similar to the percentage found normally in the Chironomus plumosus without exposure to contamination allowing to conclude that the lake was not on contamination stress and their quality must be good.
Abstract
The abstract must be reformulated because its to descriptive and should be express the results obtained without repeating the sentences found in the discussion.
Introduction
L 40-43. The sentence needed to be rephrased because I think that is not well built. I cannot understand what the authors intent to say.
L 44-47. What do you mean by “freshwater ecosystem bioassessment”. The ideia present in these lines needed to be turn out clear.
L 54-60. I feel that it is necessary to add a reference to this paragraph.
L 61-74. The argument used is good but requires to be better integrated. There are already OECD standard ecotoxicological bioassays using chironomous riparious in the evaluation of the effects of contaminants. In my opinion you have to integrate this information with the methodology proposed by you with strongs arguments that turns your methodology better.
L 75. Don’t start the sentence with “in brief”. In fact, I think that you must develop the idea more extensively.
M&M
L 91. “Zootechnics” Sorry I’m not familiar with this type of source of pollution could you explain in what it consists. Probably turn it clear to other readers would be also important by adding that to the manuscript.
L 97. You collect samples from the littoral from only one area. I understand the bathymetric zone of 4-5m depth but how could you ensure that the samples you are taking are representative of all littoral zone? Notice that you collect from 3 different areas in the central zone?
L 103. Another question that raised my concern and, in my opinion, can influence the results that you obtained were the number of samplings were completely different in the central zone and in the littoral area.
L106-110. In the continuation of the above comment another important issue besides the number of samples collected being different, the sampling effort translated in area of sampling were also different in both zones? Why is that? You must have the same number of samples and with the same area so that you can compare the results. With this methodology I don’t think that your data is comparable.
L 113 Mouthparts deformities. I think that you must add a picture to illustrate the different deformities.
L 172 I don’t understand how you obtain 7.22%
Discussion
L249. The way you wrote this sentence …”One of the Littoral zones” it looked that you analyze several littoral zones, what I understand from the M&M description is that only one area (composition of adjacent areas) but not separated. Was one littoral area or it was more?
L 269-271. I think that this sentence, in the way I understand it is dangerous. Extrapolating that the mentum deformities did not cause alterations of the head size in the deformed specimens and that their metabolic activity can be compared with normal specimens? Did you measure a battery of biochemical and enzymatic biomarkers stress to claim so peremptory that the deformities do not have any repercussion in the metabolic activity of the organisms?
L 292-294. I would complement the information with other tools…
Author Response
This research investigation is about the evaluation of the freshwater Lake Trasimeno, Italy. The authors collected several samples of macroinvertebrates in the central area and in the margin during more than 1 year. The deformity found in the Chironomus plumosus mentum was about 7% similar to the percentage found normally in the Chironomus plumosus without exposure to contamination allowing to conclude that the lake was not on contamination stress and their quality must be good.
Abstract
The abstract must be reformulated because its to descriptive and should be express the results obtained without repeating the sentences found in the discussion.
In accordance with your suggestion, we rewrote the abstract by integrating the text with the main results of the study, among those that were not reported in the previous version (max 200 words allowed).
Introduction
L 40-43. The sentence needed to be rephrased because I think that is not well built. I cannot understand what the authors intent to say.
In this regard we rewrote the sentence to clarify and simplify the meaning of the statements.
L 44-47. What do you mean by “freshwater ecosystem bioassessment”. The ideia present in these lines needed to be turn out clear.
For “freshwater ecosystem bioassessment” we mean an assessment of freshwater ecosystems through a methodology based on biological analyses. However, to better clarify its meaning, we rewrote the text.
L 54-60. I feel that it is necessary to add a reference to this paragraph.
In this regard, we rewrote the text and added references.
L 61-74. The argument used is good but requires to be better integrated. There are already OECD standard ecotoxicological bioassays using chironomous riparious in the evaluation of the effects of contaminants. In my opinion you have to integrate this information with the methodology proposed by you with strongs arguments that turns your methodology better.
In this regard, we integrated the text with a sentence on the use of some species of Chironomus in ecotoxicological bioassays as test organisms for the ecological risk evaluation.
L 75. Don’t start the sentence with “in brief”. In fact, I think that you must develop the idea more extensively.
We deleted the words “in brief” and rewrote the sentence.
M&M
L 91. “Zootechnics” Sorry I’m not familiar with this type of source of pollution could you explain in what it consists. Probably turn it clear to other readers would be also important by adding that to the manuscript.
For clarity, we replaced the term " zootechnics" with “livestock farms”.
L 97. You collect samples from the littoral from only one area. I understand the bathymetric zone of 4-5m depth but how could you ensure that the samples you are taking are representative of all littoral zone? Notice that you collect from 3 different areas in the central zone?
In the text we included this sentence to clarify the selection of the sampling plan “The current sampling design is based on our previous investigation carried out from 2000 to 2010 at Lake Trasimeno."
In particular, the littoral sampling is not on a single site but it involves an area where we identified six sites representatives of the main types of the littoral bottoms of the lake. The monitored littoral zone is located near Castiglione del Lago. This area is among the littoral zones most subject to the anthropic impact at Lake Trasimeno.
The sentence has been rewritten: “six sampling sites were selected along a 350 m area, representative of the main bottom types (sandy, silty, and macrophyte covered) present al Lake Trasimeno, situated in the coast line (1.5 m average depth), close to the town of Castiglione del Lago.”
The central zone sampling is based on “three sampling sites were selected in the same bathymetric zone (4-5 m depth)”. They are representative of the pelagic area of Lake Trasimeno which presents conditions of high homogeneity with regard to sediments.
L 103. Another question that raised my concern and, in my opinion, can influence the results that you obtained were the number of samplings were completely different in the central zone and in the littoral area.
The different number of samplings between the littoral (34) and central (17) zones does not influence the results, that are processed in relation to the different seasonal cohorts. In fact, each one of those (winter, spring, and summer cohorts) both in the central and in the littoral zones, is significantly represented by an adequate number of samplings.
L106-110. In the continuation of the above comment another important issue besides the number of samples collected being different, the sampling effort translated in area of sampling were also different in both zones? Why is that? You must have the same number of samples and with the same area so that you can compare the results. With this methodology I don’t think that your data is comparable.
The different sampling effort carried out on the central and littoral zones of the lake did not influence the study results, because the deformity incidence was processed, again in relation to the different seasonal cohorts, on a high number of larval specimens, both in the central and in the littoral zones
L 113 Mouthparts deformities. I think that you must add a picture to illustrate the different deformities.
In order to avoid doubts about subjective identifications, we took into consideration the suggestion to insert a synoptic image that summarizes and clarifies the different deformity cases analyzed.
L 172 I don’t understand how you obtain 7.22%
This value indicates the extent of the overall deformity index of the lake ecosystem, considering both the deformities of the littoral zone and those of the central zone.
Discussion
The way you wrote this sentence …”One of the Littoral zones” it looked that you analyze several littoral zones, what I understand from the M&M description is that only one area (composition of adjacent areas) but not separated. Was one littoral area or it was more?
It is a single littoral area. In particular, the littoral zone sampling was not carried out in a single site but along an area where we identified six sites, representative of the main types of bottom types of the lake (sandy, silty, and macrophyte covered).
L 269-271. I think that this sentence, in the way I understand it is dangerous. Extrapolating that the mentum deformities did not cause alterations of the head size in the deformed specimens and that their metabolic activity can be compared with normal specimens? Did you measure a battery of biochemical and enzymatic biomarkers stress to claim so peremptory that the deformities do not have any repercussion in the metabolic activity of the organisms?
According to your suggestion we replaced “metabolic activity” with “larval growth”.
L 292-294. I would complement the information with other tools…
In this regard, we rewrote the text of the sentence.
Finally, we revised the English text
Round 2
Reviewer 1 Report
Overall, the manuscript has much improved from the previous version. I would recommend another set of edits to the introduction and discussion for clarity and grammar. Otherwise i rather enjoyed the piece.
Introduction is much improved. There are still a couple of partial paragraphs that should be expanded or merged and some statements need to be more clearly stated with regards to existing literature findings and their associations to the study
Line 70: please specify the associations that the studies have found, not just that associations were found in general
Line 85-86: This statement still needs direct support
Line 128: perhaps indicate if this refers to primarily family or genera, I presume this is family from the results
Some text could be introduced in the intro on the use of family level taxonomy in biomonitoring. This is just a suggestion, but would help justify the level of identification and the implications on inferring ecological status from such data.
Line 201-202: Can you repeat the test name used and any statistics with this to just nail this on the head.
Figure 2: this is a nice visual. Is there anyway to highlight the differences among the different classes (e.g. circles). This might over clutter the figure, but would help the reader see the differences the authors are using.
Figure 3: nice plot. As a suggestion it might be easier to show differences using a stack bar plot, but i do appreciate the error bars. Could you please clarify the error bar method used (this is usually given in the help file of the associated r-function)
292-295: please check grammar
287-312: consider revising this section to form a single paragraph to summarize the main finding then segue into the first main discussion point in a subsequent paragraph.
Author Response
REVIEWER 1
Comments and Suggestions for Authors
Overall, the manuscript has much improved from the previous version. I would recommend another set of edits to the introduction and discussion for clarity and grammar. Otherwise i rather enjoyed the piece. Introduction is much improved. There are still a couple of partial paragraphs that should be expanded or merged and some statements need to be more clearly stated with regards to existing literature findings and their associations to the study.
We considered these suggestions and changed the sentences indicated in the following points and we have reviewed the text according to the referees’ remarks. The last Author of the paper, Alì Arshad is a leading personality in the environmental field at University of Florida. He laid out the second version of the paper in collaboration with us and he checked the present new version. We hope this emended text could meet the referees’ requests.
1- Line 70: please specify the associations that the studies have found, not just that associations were found in general
We specified the associations found in several studies adding to the sentence “metals, such as As, Cd, Cr, Cu, Hg, Mn, Ni, Pb, Zn, and organic compounds, such as PCBs, pesticides”. The following sentence reports the relative references.
2- Line 85-86: This statement still needs direct support
We added the references “[4]” and “[22]” to support this statement
[4] Vermeulen, A.C. Elaboration of chironomid deformities as bioindicators of toxic sediment stress: the potential application of mixture toxicity concepts. Ann Zool Fennici 1995, 32, 265-285.
[22] Di Veroli A.; Goretti, E.; Leόn Paumen, M.; Kraak, M.H.S.; Admiraal, W. Mouthpart deformities in Chironomus riparius larvae exposed to toxicants. Environ Pollut 2012, 166, 212-217.
3- Line 128: perhaps indicate if this refers to primarily family or genera, I presume this is family from the results. Some text could be introduced in the intro on the use of family level taxonomy in biomonitoring. This is just a suggestion, but would help justify the level of identification and the implications on inferring ecological status from such data.
We changed the sentence, specifying the identification level: family for macrobenthos and genus or species for Chironomids). The new sentences are: “The main taxa composing the benthic community of each sample were firstly identified to the family level using suitable taxonomic keys [57]. Chironomid specimens were subsequently identified to genus or species (Chironomus plumosus) level through the observation of peculiar features of their head capsules [58-61].” Moreover, in consideration of the possibility given, we prefer not to burden the Introduction section with the use in biomonitoring of the family taxonomical level, as the taxon object of the study on morphological deformities has always been identified at species level (Chironomus plumosus), and only the remaining benthic community was identified at family level.
4- Line 201-202: Can you repeat the test name used and any statistics with this to just nail this on the head.
We added the name of the statistical test used to support this statement. The new sentence is: “Box-Ljung test revealed no temporal autocorrelation within deformity classes among samplings”
5- Figure 2: this is a nice visual. Is there anyway to highlight the differences among the different classes (e.g. circles). This might over clutter the figure, but would help the reader see the differences the authors are using.
We modified the figure 2 and the relative caption according to the reviewer suggestions.
6- Figure 3: nice plot. As a suggestion it might be easier to show differences using a stack bar plot, but i do appreciate the error bars. Could you please clarify the error bar method used (this is usually given in the help file of the associated r-function)
We modified the figure 3 caption according to the reviewer suggestions: “Figure 3. Boxplot of C. plumosus specimens (N, log scale) in seasonal cohorts and overall 2018-2019 survey, grouped by deformity classes: CL. 1 (green, no deformity), CL. 2 (yellow, weak deformity), CL. 3 (red, strong deformity), and CL. (2+3) (purple). a) Littoral zone; b) Central zone. Box represents the interquartile range, thick lines the median, whiskers the minimum and maximum values and dots the outliers.”
7- 292-295: please check grammar
We re-wrote this sentence
8- 287-312: consider revising this section to form a single paragraph to summarize the main finding then segue into the first main discussion point in a subsequent paragraph.
In accordance with the reviewer suggestions, we revised the text of these sentences and merged some paragraphs.
Reviewer 3 Report
I would like to suggest that the author's would reply to the authors with more complete responses than just 3 or 4 lines in future interactions with reviewers.
Author Response
REVIEWER 3
I would like to suggest that the author's would reply to the authors with more complete responses than just 3 or 4 lines in future interactions with reviewers.
We thank the reviewer and will keep this suggestion well in mind in future works.